# A Rapid Review of Interventions to Improve Care for People Who Are Medically Underserved with Multiple Sclerosis, Diabetic Retinopathy, and Lung Cancer

**DOI:** 10.3390/ijerph21050529

**Published:** 2024-04-24

**Authors:** Sarah Mossburg, Mona Kilany, Kimberly Jinnett, Charlene Nguyen, Elena Soles, Drew Wood-Palmer, Marwa Aly

**Affiliations:** 1American Institutes for Research, Arlington, VA 22202, USA; 2Department of Social and Behavioral Sciences, UCSF Institute for Health and Aging, San Francisco, CA 94158, USA; 3Department of Applied Health Sciences, School of Public Health, Indiana University Bloomington, Bloomington, IN 47405, USA

**Keywords:** rapid review, interventions, medically underserved

## Abstract

In the United States, patients with chronic conditions experience disparities in health outcomes across the care continuum. Among patients with multiple sclerosis, diabetic retinopathy, and lung cancer, there is a lack of evidence summarizing interventions to improve care and decrease these disparities. The aim of this rapid literature review was to identify interventions among patients with these chronic conditions to improve health and reduce disparities in screening, diagnosis, access to treatment and specialists, adherence, and retention in care. Using structured search terms in PubMed and Web of Science, we completed a rapid review of studies published in the prior five years conducted in the United States on our subject of focus. We screened the retrieved articles for inclusion and extracted data using a standard spreadsheet. The data were synthesized across clinical conditions and summarized. Screening was the most common point in the care continuum with documented interventions. Most studies we identified addressed interventions for patients with lung cancer, with half as many studies identified for patients with diabetic retinopathy, and few studies identified for patients with multiple sclerosis. Almost two-thirds of the studies focused on patients who identify as Black, Indigenous, or people of color. Interventions with evidence evaluating implementation in multiple conditions included telemedicine, mobile clinics, and insurance subsidies, or expansion. Despite documented disparities and a focus on health equity, a paucity of evidence exists on interventions that improve health outcomes among patients who are medically underserved with multiple sclerosis, diabetic retinopathy, and lung cancer.

## 1. Introduction

The U.S. Department of Health and Human Services defines health equity as the attainment of the highest level of health for all people [1]. Since the late 1990s, a growing body of health equity research has increased our understanding of the impacts of economic and social conditions on health [2]. The scope of health equity initiatives spans community-based projects led by non-profit organizations to national health programs co-ordinated by federal agencies. Despite recent increases in initiatives intended to advance health equity, disparities in health and healthcare persist.

Many factors are associated with disparities in health outcomes. Socioeconomic status, geographic location, and race and ethnicity are associated with lower access to care, poor treatment, and negative health outcomes in the United States. Social determinants of health (SDOH) categorize the long-standing structural and systemic inequities that reinforce health disparities [3,4]. SDOH include the context in which people live and can be grouped into five domains: economic stability, education access and quality, healthcare access and quality, neighborhood and built environment, and social and community context [5]. Examples of SDOH include health literacy, social cohesion, and discrimination. These factors cause some groups of people to experience higher rates of illness and death across various health conditions [4].

In addition, the effects of health disparities have broader implications for our national economy. The research shows that health disparities are costly, amounting to an approximately $93 billion excess in medical care costs annually [4]. Chronic diseases account for a significant percentage of annual healthcare spending, with approximately 75% of the national health expenditure used to treat patients with one or more chronic diseases [6]. Chronic diseases are typically characterized by their longevity, lasting more than three months, and worsening over time [7]. In the US, three chronic conditions, heart disease, cancer, and diabetes, account for the top causes of death, disability, and healthcare costs [8].

Health disparities exist all along the care continuum, which encompasses a patient’s access to and experience with healthcare, including screening, diagnosis, access to treatment and specialists, adherence, and retention in care. Chronic conditions with health disparities across the care continuum include multiple sclerosis, diabetic retinopathy, and lung cancer. For people who have a chronic condition and are medically underserved, obtaining the needed care is even more complicated. People who are medically underserved may be those that live in a geographic location without sufficient resources to manage health needs, they may lack resources to access care, or they may be from a historically marginalized population that has faced social and economic barriers to care. For people with multiple sclerosis, accessing disease-modifying therapies (DMTs), especially for individuals with relapsing–remitting multiple sclerosis, can be difficult because of the high price, which then affects insurance, pharmacy, or provider policies [9]. For people with diabetic retinopathy, obtaining the diabetic eye exams necessary for diagnosis is difficult without routine access to specialists such as an ophthalmologist. Finally, Americans who identify as Black, Indigenous, and people of color (BIPOC) who are diagnosed with lung cancer experience worse health outcomes than those who identify as White. They are more likely to be diagnosed at a later stage of disease, less likely to receive surgical treatment, and more likely not to receive any treatment [10].

The U.S. healthcare system often incentivizes patients and providers to restrict healthcare utilization to decrease costs. Unfortunately, these incentives create barriers which often lead to increased costs in the form of lost productivity and wages, and pain and suffering for patients [11]. Interventions aimed at addressing barriers experienced by patients with chronic conditions may improve outcomes like medication adherence, improve access to screening and treatment, and improve health outcomes, while reducing the economic strain on payers, patients, and the healthcare system. To effect health disparities, interventions may focus on making changes at one or more levels, from the individual patient level up to the healthcare system level [12]. Depending on the level, interventions may focus on changing knowledge or behaviors (patient or provider), operational policies or practices (organizational), or large-scale policies (system). Addressing health disparities likely requires interventions at one or more levels, but a systematic search for interventions across the care continuum and level of intervention does not currently exist.

Given the documented disparities experienced by patients with chronic conditions and a growing shift toward providing more equitable care, this paper seeks to identify evidence-based interventions to improve health and reduce disparities in treatment access, quality of care, or health outcomes. Our team had recently conducted an analysis of the geographic distribution and sociodemographic characteristics of providers for three clinical areas: neurology, ophthalmology, and lung cancer specialists [13]. As a companion to that work, we limited our search to multiple sclerosis, diabetic retinopathy, and lung cancer, anticipating the possible applicability of the evidence across conditions. The following research questions focused on identifying interventions and determining their effectiveness:Among patients with multiple sclerosis/diabetic retinopathy/lung cancer, what kind of interventions have sought to address any point in the care continuum in order to improve health and reduce disparities in screening, diagnosis, access to treatment and specialists, adherence, and retention in care?Among patients with multiple sclerosis/diabetic retinopathy/lung cancer, what is the effectiveness of interventions that address any point in the care continuum to improve health and reduce disparities in screening, diagnosis, access to treatment and specialists, adherence, and retention in care?

## 2. Methods

A rapid literature review of evidence-based interventions to improve health and reduce disparities was conducted among groups of people who are medically underserved with multiple sclerosis, diabetic retinopathy, and lung cancer [14]. For the purposes of this review, medically underserved people included people with low income, who are uninsured and underinsured, without a usual source of care, living in rural areas, or who identify as BIPOC. The Preferred Reporting Items for Systematic Reviews (PRISMA) guidelines for reporting and conducting systematic reviews [15] were followed. The Population Intervention Comparison Outcome Time (PICOT) framework was used to help guide the review strategy [16].

### 2.1. Inclusion and Exclusion Criteria

Eligibility criteria were based on the PICOT question, and the search was limited to studies published within the previous 5 years, conducted in the United States, written in English, and included patients older than age 18. Eligible study designs included randomized controlled trials (RCTs), observational studies, qualitative evaluations, mixed-methods designs, and program evaluations. These criteria were selected to identify the most current evidence-based interventions and findings among people who are medically underserved.

### 2.2. Search Strategy

PubMed and Web of Science databases were searched for this review. A health librarian provided input on developing and testing our search strategy, including which databases to select given the rapid timeline which limited our ability to search all relevant databases. A combination of MESH terms and keywords were used to capture indexed and nonindexed Medline articles as well as other nonindexed articles from non-Medline databases available in PubMed. Example MESH terms used in our search include “Multiple Sclerosis” [Mesh], “Lung Neoplasms” [Mesh], “Macular Edema” [Mesh], “Diabetic Retinopathy” [Mesh], “Medically Uninsured” [Mesh], and “Health Disparity, Minority and Vulnerable Populations” [Mesh]. A full list of MESH terms and keywords is available in Appendix A.

The structured search included terms for three health conditions (multiple sclerosis, diabetic retinopathy, and lung cancer), people who are medically underserved, and the patient care continuum. The care continuum encompasses a patient’s access and experience with healthcare, including screening, diagnosis, access to treatment and specialists, adherence, and retention in care. Groups of people who may be medically underserved include people who identify as BIPOC, live in rural areas, have low incomes, are uninsured or underinsured, or lack usual care or a medical home [17]. Before conducting the searches, search terms were tested, and input was sought from additional clinical experts in the field.

Three separate searches were conducted in PubMed and Web of Science, one for each clinical condition combined with the search terms for people who are medically underserved and the patient care continuum. After removing duplicates, the searches yielded 4056 articles (Figure 1). We used Rayyan, an online collaborative platform, to manage the selection and screening process of relevant studies. One reviewer screened each article title and abstract based on inclusion criteria for eligibility. A second reviewer examined a sample (<10%) of randomly selected excluded articles to confirm agreement. Title and abstract screening yielded 228 articles for inclusion at this stage. For full-text screening, two reviewers independently reviewed each study for inclusion. Reviewers discussed conflicting decisions to reach a consensus. The full research team discussed unresolved cases until a consensus was reached. To ensure agreement, a third reviewer evaluated a sample (<10%) of randomly selected excluded articles. Full-text screening yielded a final sample of 44 articles: five for multiple sclerosis, 12 for diabetic retinopathy, and 27 for lung cancer.

### 2.3. Data Extraction

To standardize extraction, a data dictionary and data extraction spreadsheet were created. Data were extracted from eligible studies into the shared spreadsheet. Data extraction elements based on the PICOT framework included information on study design, participants, setting, intervention, outcomes, key findings, and limitations. The TIDieR checklist guided decisions about extraction of intervention data elements [18]. For the first 17 studies, two reviewers independently extracted data from each study. A third reviewer then evaluated these extractions to ensure consensus and provided feedback. A single reviewer extracted data from the remaining studies, and a second reviewer independently confirmed agreement. The full team discussed all disagreements until a consensus was reached.

### 2.4. Quality Assessment and Data Synthesis

Researchers assessed the quality of studies using the National Institutes of Health National Health, Lung, and Blood Institute [19] study quality assessment tools; for designs for which there was no NHLBI quality assessment tool, JBI critical appraisal tools were used [20]. Descriptive statistics of the studies extracted across all three clinical condition were summarized. Tables to analyze and summarize the data extracted for the review were created and a narrative synthesis of the findings by clinical condition was conducted.

## 3. Results

Table 1 presents the number of studies identified across each point in the care continuum by level of intervention. Our search did not yield articles focused on retention in care for any clinical condition. Across all clinical conditions, 13 studies focused on patient-level interventions, 17 focused on organization-level interventions, and 13 focused on system-level interventions. Screening was the most common point in the care continuum for diabetic retinopathy and lung cancer; no studies addressed screening for multiple sclerosis. The body of evidence for patients with multiple sclerosis was limited and focused primarily on access and adherence to treatment. Table 2 presents the types of interventions identified by the care continuum and level of interventions across the clinical conditions.

Among the studies we identified, two-thirds (*n* = 29) included patients who identify as BIPOC. Moreover, 26% of studies included patients in rural settings, 19% included patients with low incomes, and 12% included patients who were underinsured or uninsured.

### 3.1. Multiple Sclerosis

Five studies with interventions to improve care for people with multiple sclerosis who are medically underserved at some point along the care continuum were identified (Table 3). The Appendix A contain an expanded table with details of study designs, available participant demographics, care continuum, level of intervention, outcomes, and findings.

#### 3.1.1. Access to Treatments or Specialists

Two studies included an intervention that increased access to treatment or specialists among people with MS who are medically underserved. Hartung et al.’s [23] large retrospective cohort study (*n* = 39,661) addressed access to treatment via a cost-sharing subsidy and found that, across demographic categories, newly diagnosed Medicare beneficiaries with reduced cost-sharing via a low-income subsidy were more likely to initiate early self-administered DMT than those without a subsidy. Plow et al. [25] conducted an RCT (*n* = 208) of a patient-level intervention examining changes in physical activity among patients with multiple sclerosis using telemedicine to provide access to occupational therapists via structured teleconferences. At 12 weeks, compared to the social support intervention, the other two interventions improved physical activity. The fatigue self-management and physical activity intervention also decreased the impact of fatigue. Differences were not sustained at 24 weeks.

#### 3.1.2. Adherence to Treatment

Three studies included an intervention that addressed adherence to treatment among a specific group with disparities in access or utilization of multiple sclerosis care. All were conducted among Black or African American participants. Of these studies, two smaller (*n* < 35), single-group, pre–post feasibility studies included an intervention to affect participants’ physical activity levels [21,24]. The interventions for these studies included multiple components, such as an exercise program, behavioral coaching, and education materials. Both interventions were safe and feasible, and increased activity levels among participants.

The third study involved patient-level interventions that addressed factors that influenced medication choice among patients with multiple sclerosis. Cascione et al. [22] found higher rates of adherence at 48 weeks among those taking oral versus injectable DMTs; injection-related issues were the most commonly cited reason for stopping injectable DMTs.

### 3.2. Diabetic Retinopathy

Twelve studies with an intervention to improve screening for or diagnosis of retinopathy for patients with diabetes were identified (Table 4).

#### 3.2.1. Screening

Of the nine studies that focused on interventions to improve screening, six included teleophthalmology use in primary care, two included mobile clinics, and one included an automated phone reminder.

All three studies that evaluated changes in screening rates used a pre–post intervention design based on historical cohorts. Increases in screening rates for eligible patients ranged from 14.8% to 30.2% [29,30,32]. One study found that the intervention also reduced wait times for screening by 89.2% [29]. Completion rates for referrals among patients who screened positive for diabetic retinopathy differed (12% versus 60%) between the two studies reporting these rates [30,31]. Referral completion rates were higher among rural versus urban patients; both groups consisted entirely or substantially of patients with Medicaid.

Two studies evaluated patient perceptions of or satisfaction with teleophthalmology use in primary care. A small (*n* = 23) mixed-methods study that compared patients’ experience with teleophthalmology to experiences with a dilated eye exam or no exam found that patients generally perceived teleophthalmology to have a high value for its convenience and ability to detect disease, and were willing to pay an equivalent copay, although cost was a concern [34]. Similarly, a larger (*n* = 247) observational study using pre- and post-intervention surveys found that, overall, patients expressed satisfaction with telemedicine and preferred it to in-person visits, although researchers noted that patients who had had prior in-person exams were less likely to prefer telemedicine [36].

A final study interviewed patients and PCPs to identify barriers to and facilitators of the use of a referral, walk-in teleophthalmology program. The primary patient-level barriers identified were unfamiliarity with teleophthalmology, misconceptions about screening, and logistical difficulties; facilitators included a PCP recommendation and convenience factors. Primary PCP-level barriers included not knowing when screening was due for individual patients, and unfamiliarity with teleophthalmology; facilitators included the ease of the referral process and communication of results [32].

Two studies evaluated mobile ophthalmology clinics. Of the 6% of screened patients diagnosed with retinopathy, researchers reached 48% by phone within 4 months of screening and confirmed that 71% of patients had completed a follow-up with an ophthalmologist [28]. The second mobile clinic intervention focused on the access to screening and costs associated with a medical student-run screening clinic supervised by an ophthalmology attending and resident [35]. Between October 2013 and February 2020, the mobile clinic referred 178 patients for advanced ophthalmologic care, including glaucoma, diabetic retinopathy, and age-related macular degeneration. The free clinic was estimated to have provided 1271.3 Medicare relative value units for services, an equivalent of $119,263 overall, or $136 per screening.

The final study evaluated the use of an automated phone reminder combined with a telephone reminder by a medical assistant for patients scheduled for screening exams during the week prior to their appointment compared with a medical assistant telephone reminder alone. Adding automated reminders improved attendance rates and narrowed the disparity between African American and Latino patients, although rates for both groups remained low (51.6% and 62.3%, respectively) [33].

#### 3.2.2. Diagnosis

Of the three studies seeking to improve the diagnosis of retinopathy among patients with diabetes, two involved artificial intelligence (AI) algorithms that evaluated optical coherence tomography angiography (OCTA) or retinal images compared to a standard approach. Alam et al.’s [28] study used known images for diabetic retinopathy, sickle-cell retinopathy, and normal controls to evaluate the feasibility of an AI system to identify diabetic retinopathy. The study demonstrated the feasibility of this approach with a diagnostic performance of 97.84% sensitivity and 96.88% specificity for diagnosing disease in images with known retinopathy compared with those without disease [28]. Abramoff et al. [26] conducted a large (*n* = 900) superiority trial comparing the diagnostic performance of an AI-based system with the gold standard for diagnosing diabetic retinopathy based on nonmydriatic retinal images. The study showed that the AI system met pre-specified endpoints for superiority compared to the standard approach with a sensitivity of 87.2% and specificity of 90.7%.

The third study was an observational pilot study (*n* = 118) in which researchers compared agreement with the remote interpretation of OCT with in-person eye exams for detecting retinal abnormalities [37]. Eye doctors conducted clinical exams in mobile vans. Researchers found that, among patients with diabetes, OCT and clinical exam had moderate agreement in diagnosing retinopathy, with a third of retinopathies detected by OCT alone, a third by clinical exam alone, and a third with both.

### 3.3. Lung Cancer

We found 26 studies with an intervention to improve care along the continuum for patients with lung cancer (Table 5).

#### 3.3.1. Screening

Of the 18 studies that included interventions or programs focused on screening for lung cancer, four were observational studies of low-dose computed tomography (LDCT) screening programs [42,56,57,62].

Three non-mobile LDCT programs were feasible to implement and screened between 169 to 7807 patients. However, Erkmen et al. [42] noted challenges with ongoing screening; after 1 year, only 23.7% of those with negative screenings had followed up; at 2 years, 35.4% of those with positive screenings but no cancer had followed up. Randhawa et al. (2018) noted that physicians reported time constraints and precertification requirements as barriers to referral to the LDCT program.

Five studies included comparisons of screening methods or criteria. In a secondary data analysis of the original National Lung Screening Trial (NLST) data using a synthesized sample with a higher proportion of Black individuals than the original sample [55], compared with chest X-ray, LDCT screening had a greater reduction in lung cancer mortality among Black patients compared with the original NLST results. A retrospective cohort study (*n* = 774) comparing the use of NLST criteria plus Lung Imaging Reporting and Data Systems (Lung-RADS) categories with NLST baseline rates on subsequent testing and cancer diagnosis found that the use of Lung-RADS led to 13.3% fewer additional tests compared with NLST alone [40]. Using Lung-RADS categories increased the percentage of patients diagnosed with cancer among those patients who had additional testing by five times compared with diagnosis rates among the original NLST population who had had additional testing.

Two large observational retrospective studies (*n* = 530 and *n* = 980) compared the use of National Comprehensive Care Network (NCCN) screening criteria with 2013 United States Preventive Services Task Force (USPSTF) screening guidelines on screening eligibility rates. Both studies retrospectively evaluated the screening criteria among patients with known lung cancer to determine whether patients would have met screening criteria before diagnosis. Olazagasti et al. [50] found that significantly more Hispanic/Latinx patients did not qualify for screening based on USPSTF guidelines; eligibility rates did not differ between African Americans and those who were White, Asian, or other races when comparing the two guidelines. Thurlapati et al. [61] found that one-third of patients with lung cancer did not meet the 2013 USPSTF guidelines, and, of those, only 12.5% would have been eligible for screening based on revised NCCN guidelines using an individual risk-based calculator. Among those who would not have met eligibility criteria, 50% were African American. Another large retrospective cohort study (*n* = 883) evaluated the sensitivity of the PLCOm2012 model versus USPTSF guidelines for lung cancer screening. It found that, among African American patients, the PLCOm2012 model had greater sensitivity for lung cancer screening [53].

Three studies evaluated screening decision aid use on knowledge, decisional conflict, decision regret, and acceptability, among other outcomes. In a moderate-sized (*n* = 74) pre–post use evaluation, researchers found that the use of shouldiscreen.com led to small improvements in knowledge and increased concordance with recommendations [46]. In a large (*n* = 237) RCT comparing shouldiscreen.com with Option Grids, another decision aid, researchers found that patients using Option Grids had less decision regret and greater knowledge regarding next steps for positive screenings and the potential need for invasive procedures (*p* = 0.0198 and *p* = 0.02, respectively) [59]. Finally, in an evaluation of an adapted Agency for Healthcare Research and Quality (AHRQ) lung cancer screening decision aid, “Is Lung Cancer Screening Right for Me?” (*n* = 50), researchers found, among older Chinese Americans, that, although participants had good knowledge after using the aid, 66.7% were unable to understand the content without help [47].

Three studies evaluated patient education interventions. Education delivery differed slightly among two studies evaluating the same program in different locations (one education session versus four sessions). Both programs showed some improvements in knowledge about lung cancer screening guidelines among participants [63,64]. The third study compared two education seminars and found that seminars developed for Cantonese-speaking Chinese Americans had an impact on the beliefs and stated behaviors of Chinese Americans, although, at baseline, more than two-thirds of participants were aware that screening tests for lung cancer were available. Both groups had high knowledge about cancer prevention at baseline. Changes in knowledge, attitudes, and screening intent were minimal between the groups [44].

One large RCT (*n* = 1200) that included patients with low incomes found the proportion of patients who received screening via a CT scan among patients randomized to a patient navigator was higher compared with those who received usual care [54].

The final two studies focusing on screening included a quality improvement (QI) project and a lung cancer screening awareness campaign. In a large QI project (*n* = 544) using a weekly report of eligible patients, a paper reminder to physicians, and provider notifications for missed screenings, screening rates among rural patients increased to 85% of eligible patients compared to 68% prior to the intervention [58]. A lung cancer screening awareness campaign in rural Michigan was unable to show differences in lung cancer screening rates between patients with more than and less than a 30-year smoking history [60].

#### 3.3.2. Access to Specialists or Treatment

Two studies included an intervention to provide access to specialist care, while four addressed access to treatment. A large (*n* = 956) observational case–control study evaluated the use of a molecular tumor board on overall survival [45]. Compared to those with reviews, those without reviews had poorer survival (HR = 8.61; 95% CI: 3.83, 19.31). A qualitative analysis of chart notes from an RCT explored the benefits of early outpatient palliative care consultation for people with newly diagnosed advanced cancer [38]. Early consultations (within 60 days) addressed patient and family concerns that may not have been addressed in typical cancer care visits.

Two studies of interventions to improve access to care included qualitative studies of the same mHealth app. A small (*n* = 19) qualitative analysis of barriers and facilitators of Breathe Easier, an mHealth app with content for mindfulness-based cancer recovery, found that the primary benefits were convenience and credible information, while the top concerns included cost and difficulty of use [39]. Another small study (*n* = 12) that assessed the cultural acceptability of the Breathe Easier app among African American patients and families found that the app was well-understood and would benefit survivors, but participants raised concerns about health literacy problems [52].

The final two studies of interventions to improve access to care included a multifaceted QI intervention, and state-wide insurance expansion. A large (*n* = 360) pragmatic trial using retrospective controls evaluated a system-based intervention to reduce disparities in treatment among patients with lung cancer. It found that Black and White patients in the intervention group had a similar receipt of curative treatment, whereas Black patients in the retrospective group had lower rates of receipt of curative treatment compared with White patients [64]. A large (*n* = 67,987) quasi-experimental study that evaluated the impact of state-level Medicaid expansion on the utilization of high-volume hospitals among patients with cancer found that, while rates of complex surgical care increased by 14% relative to non-expansion states, the probability of undergoing surgical resection at high-volume hospitals did not change [49].

#### 3.3.3. Diagnosis

Another large (*n* = 101,227) retrospective study looked at the effect of state-level Medicaid expansion on early-stage cancer diagnosis and 2-year survival compared with non-expansion states [48]. Researchers found that outcomes did not differ for women in states that did and did not expand Medicaid. Among men, compared with those in states that did not expand Medicaid, those in expansion states had greater increases in 2-year survival and early-stage diagnosis.

#### 3.3.4. Patient Adherence to Treatment

A secondary data analysis of a large (*n* = 137) RCT evaluating the effects of having a companion present during care conversations with oncologists found that, when a companion was present, oncologists provided more patient-centered communication and spent more time with patients [51]. Oncologists also perceived patients to be more active participants and to have more social support compared with when a companion was not present.

## 4. Discussion

Although our search yielded a fair number of studies across the clinical conditions, in general, we found limited interventions focused on SDOH-based root causes of disparities, such as the conditions in which people live, compositions of social networks, structural and institutional racism, and the nature of social relations. Most SDOH-based examples were in lung cancer care. To broadly identify any interventions with the published evidence, the search was agnostic to the level of intervention and included social, technical, or combined interventions. Among the studies identified, levels of intervention were characterized as patient-, provider-, organization-, or system-level; no studies solely addressing provider-level interventions were found. Several studies characterized as organization-level interventions were multicomponent and included provider-level elements. The design of complex interventions may include multi-level elements, allowing for interactions among elements. For example, an intervention with provider education components and organization-wide system change may offer more benefit as the two components reinforce each other. Below, we discuss the themes that emerged in interventions to address SDOH across the interventions, followed by the remaining gaps.

### 4.1. Addressing Geographic Barriers to Care

Among rural patients, geographical context is often a barrier to care; therefore, interventions to improve access to specialists and integrated care could help improve outcomes. Integrated care centers have been shown to improve survival outcomes among patients with lung cancer, potentially because of improved access to newer therapies [65]. It is possible that this is influencing the successful outcomes of the study of a molecular tumor board in Kentucky [45].

Studies also addressed geographical barriers via telemedicine. Several studies deployed teleophthalmology in primary care practices to improve diabetic retinopathy screening, in rural areas and nonrural areas. Disparities in broadband access among rural areas may decrease the generalizability of such interventions. This may not be an issue in using teleophthalmology deployed at primary care sites.

In some instances of home-based interventions, telephone-delivered interventions may provide an alternative to telemedicine that circumvents limited broadband access issues. Plow et al. [25] showed partial support for their intervention using teleconferences with occupational therapists and tailored phone calls to increase physical activity and reduce fatigue. Notably, however, improvements were not sustained at follow-up.

### 4.2. Improving Health Literacy

Health literacy is an SDOH that may contribute to outcome disparities. High health literacy can help people understand health information to make well-informed decisions. People with low health literacy may have difficulty decoding screening guidelines to ascertain their personal risk for disease and potential benefit from screening. We identified studies that could be considered to impact health literacy by increasing knowledge of screening. In their evaluation of a web-based decision aid for lung cancer patients culturally adapted for use with African Americans, Lau et al. [46] found low concordance between a preference for screening and guideline recommendations among those who used the aid. The authors suggest that this may be related to the challenges of interpreting the harms and benefits of screening. This finding may reflect the limited impact of the decision aid on improving health literacy. In the United States, people from different cultural backgrounds and who do not speak English as a first language experience some of the greatest health literacy disparities [66]. In the only study we found of an adapted decision aid for use among groups with cultural and language barriers, authors noted that most participants still reported needing help to understand the content [47].

### 4.3. Improving Social Cohesion

Another SDOH that may be relevant to reducing disparities and which has been shown to decrease mortality is social cohesion, an indicator of the strength of relationships among community members [67]. Although no studies were designed with social interventions intended to improve health, some included components that address social cohesion. Otto et al.’s [51] study found that oncologists perceived Black patients with lung cancer to have greater social support and be more active participants when a companion was present. No conclusions could be made about whether these perceptions changed communication patterns; however, when a companion was present, oncologists provided more patient-centered communication and spent more time with patients. Erkmen et al.’s [43] community-engaged screening program attempted to leverage social networks to spread health behavior, identify people who would benefit from screening, and encourage lung cancer screening. The study engaged 505 people in screening but did not improve follow-up adherence to annual screenings for those who would have benefited from them.

### 4.4. Decreasing Discrimination

Individual and structural discrimination may contribute to disparities in outcomes among people who identify as BIPOC. Interventions addressing unintentional individual bias may help improve outcomes, although evidence is limited. An intervention to enhance racial equity in cancer treatment, including explicitly aggregating treatment completion rates by race, reduced disparities in outcomes, producing similar treatment completion rates among Black and White patients [41,42]. The intervention also included the use of nurse navigators, which has a documented effectiveness at improving care among patients with other cancers [68].

The potential bias in screening guidelines is unclear. Such bias may be associated with structural discrimination which are processes or conditions that limit the opportunities, resources, and well-being of specific groups on a large scale such as at the community, state, or national level [69]. The appropriate identification of who should be screened for lung cancer is important to identify cancer at early stages while minimizing over-testing and the burden and stress of false-positive results. False-positive tests can lead to invasive procedures and an associated risk of harm. African American patients have been historically underrepresented in trials determining the cancer risk factors upon which screening guidelines are based. The lack of representation in these trials could inadvertently lead to bias in screening guidelines.

We found several studies that compared screening methods to produce a better lung cancer risk determination among people who identify as BIPOC [50,53,61]. Alternative risk prediction models may more accurately identify Black or African American patients at increased risk for lung cancer [53,61].

### 4.5. Affordability of Care

Affordability of care is an important social factor in access and treatment that is necessary but not sufficient to improve outcomes. Three studies evaluated the effect of Medicaid or insurance expansion on outcomes with mixed results. While cost sharing effectively increased the likelihood of initiating early self-administered DMTs [23], Medicaid expansion only had a differential effect by sex on 2-year survival and early-stage diagnosis [48] and did not change the probability of surgical resection for lung cancer care [49]. These mixed findings may reflect that cost is only one aspect in the complex interplay of factors affecting care accessibility. Patients may prefer to continue existing care relationships, prefer the convenience of local services, lack awareness of the availability of other services, or be unable to travel greater distances for care at high-volume hospitals.

### 4.6. Remaining Gaps

The relative paucity of studies identified in this review evaluating interventions addressing social needs to improve care among medically underserved patients with multiple sclerosis, diabetic retinopathy, and lung cancer may be a reflection of the current general lack of health equity interventions. Figueroa et al. [70] found that, despite similar social needs screening rates, rural, critical access, and safety-net hospitals all reported fewer community partnerships to address SDOH. Rural and critical access hospitals reported fewer programs or interventions to address SDOH. While there is acknowledgement among healthcare leaders about the foundational nature of health equity to improve outcomes, the field has not yet progressed towards developing, testing, and implementing interventions to improve equity. Although 65% of community health needs assessments performed by urban hospitals required to retain tax-exempt status included at least one health equity term, only 9% included an explicit health equity activity [71].

Social risk screening is emerging as a low-effort first step to identify people who would benefit from referral to social services [72,73]. Many programs report on the identification of needs, but there is mixed reporting on patient uptake of services [74]. Currently, only about a third of those who receive referrals to a community-based organization end up receiving assistance [73]. Research is needed to better understand why patients do not receive the needed social services [75]. One reason may be related to community-based organization capacity issues, which vary depending on the type of assistance and geographic location [73].

An important ongoing challenge identified across conditions is the continued engagement in screening or follow-up care for patients who remain at high risk for developing or experiencing complications of disease. Although not discussed in most studies, a few that reported an improvement in screening rates noted challenges in follow-up care [28,31,43]. Perhaps one basic strategy for retaining patients in ongoing screening via teleophthalmology comes from Serrano et al. [36], who suggested that doctors avoid setting unrealistic expectations for teleophthalmology when offering it to patients, because they found that satisfaction with the service was influenced by expectations and experiences of disconfirmation. Social factors that prevent initial engagement in screening, such as the ability to get to appointments, cost of co-pays, inability to take time off work, lack of childcare, or lack of local services, all likely impact retention in care. Patients may be able to overcome these barriers on a limited basis for an initial screening but lack resources for continued follow-up.

### 4.7. Strengths and Limitations of the Literature Included

Because the search was broad and included a wide range of designs, the body of literature reviewed is heterogeneous and provides an overview of the evidence across the conditions that can be difficult to draw conclusions from. The small number of articles identified for multiple sclerosis limits the ability to draw meaningful conclusions about this research, and the points in the care continuum of focus for multiple sclerosis were very different from those for diabetic retinopathy or lung cancer, which focused on early identification and screening. The number of well-designed studies and similar outcomes among the diabetic retinopathy literature on the use of teleophthalmology in primary care increases confidence in the ability of this intervention to improve screening.

The results of this review should be considered in the context of a few key factors. The search strategy included the clinical conditions as a required element. The body of literature addressing disparities in outcomes across populations of people who are medically underserved may be agnostic to clinical condition or may have been tested among patients with other clinical conditions. Likewise, methods to address disparities may not have a clear intervention. For example, research shows that racial concordance among patients and providers significantly increases the likelihood of seeking preventive care and care for ongoing medical problems [76].

Further, while search terms for populations who are medically underserved were deliberately included in this research, studies were included that were not exclusively focused on a group that is medically underserved if the study included at least 20% of people who were part of a disproportionately affected group. Interventions evaluated in these studies were, therefore, less likely to specifically address social or geographical context—key risk factors for disparities.

### 4.8. Strengths and Limitations of the Present Review

This study’s use of a rapid review format enabled the quick evaluation of the evidence while adhering to methodological requisites to minimize bias. Title and abstract screening were limited to a single reviewer; however, a second reviewer randomly evaluated a small sample of articles for agreement. Similarly, to facilitate data extraction, after two reviewers extracted data from almost 40% of articles, the approach switched to a single reviewer with the validation of select elements by another reviewer.

Several limitations of this review should be considered. We used a 5-year cutoff, which may have excluded relevant older articles. The focus was on multiple broad search terms to avoid limiting findings to a narrow list of interventions, but this may have excluded keywords or MESH terms for specific interventions that authors used to tag relevant studies. In addition, points in the care continuum are constructs that are difficult to account for in a search strategy. Alternative ways to describe or define the constructs could have yielded more results. Because of differences between healthcare systems limiting the generalizability of international studies, this search focused solely on United States-based studies.

### 4.9. Future Research and Conclusions

The findings of this review point to a few potential avenues for future research to address disparities in care. There was very little evidence of interventions to improve care among people from ethnic and racial minorities, or communities that are undeserved with limited access to multiple sclerosis care. There is a need to test existing evidence-based strategies among these groups to evaluate reductions in disparities. Importantly, despite successful strategies to improve screening for diabetic retinopathy and lung cancer, notable gaps exist in ongoing screening and follow-up care for some patients. Identifying interventions to retain patients who continue to be or are at increasing levels of risk in screening regimens and care is critical to reducing disparities in outcomes. Finally, this review did not identify studies focused on workforce interventions to improve access to care, possibly because workforce interventions are not specialty- or condition-specific. Existing workforce interventions such as the ECHO model^®^, that trains primary care doctors or other healthcare workers to take on pre-screening or other efforts to extend the referral work, or the virtual hub and spoke model where specialists provide virtual consults with PCPs and other generalists, offer viable alternatives to address these shortages of specialists. Future research could focus on interventions affecting the makeup and distribution of the healthcare workforce to explore ways to influence outcomes across the care continuum.

Disparities in health outcomes are well-documented among patients with a range of clinical conditions, not just multiple sclerosis, diabetic retinopathy, and lung cancer. The causes of the disparities are complicated, and likely include complex, multifactorial issues at the patient, provider, organization, and system levels. Given this complex causal structure, it seems doubtful that simple, single-level solutions will be successful in resolving disparities; therefore, multilevel solutions may be required. However, health equity is a critically important goal to improve the well-being of all people, and, ultimately, is worth the difficult work required to achieve it. The time has come to move beyond studies documenting the existence of disparities into the hard work of addressing them and building evidence around effective practices and policies.

## Figures and Tables

**Figure 1 ijerph-21-00529-f001:**
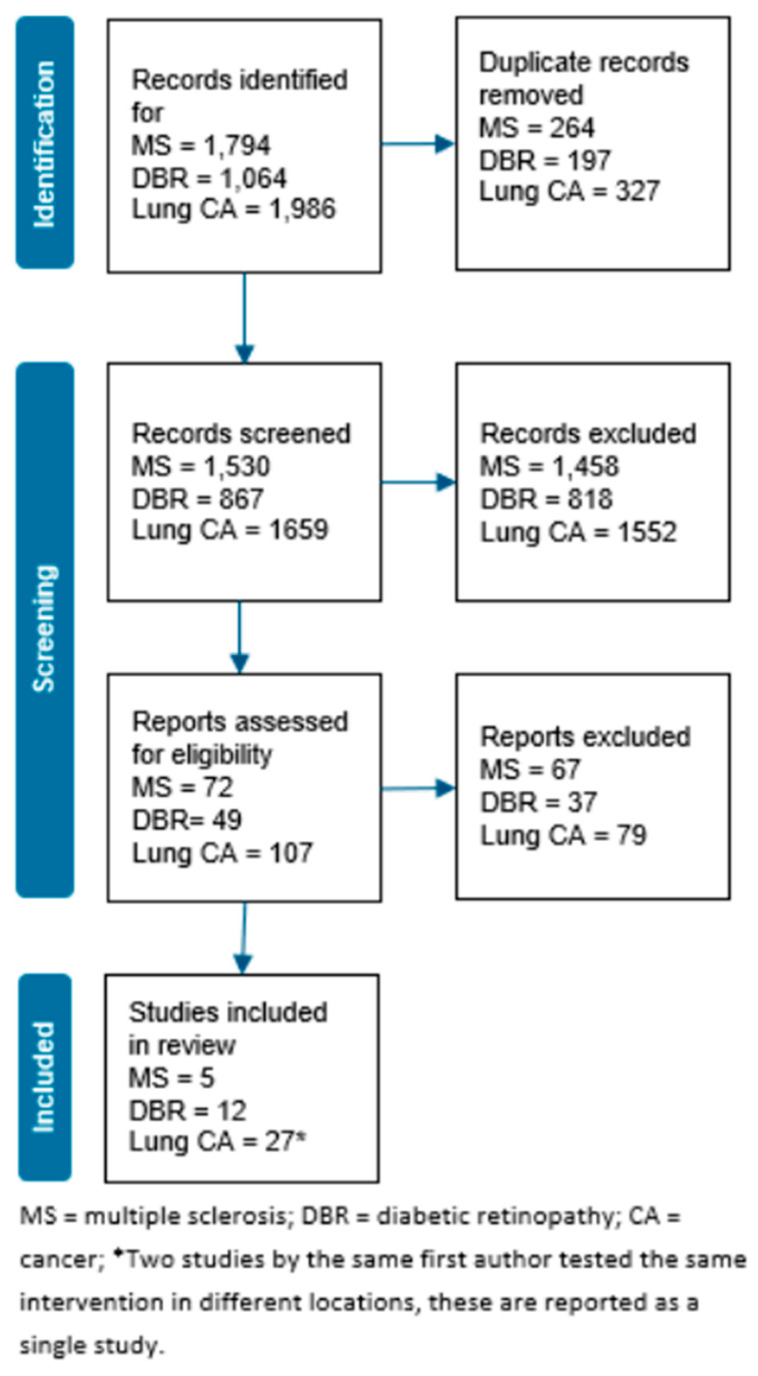
PRISMA Search Diagram.

**Table 1 ijerph-21-00529-t001:** Number of studies identified by care continuum and level of intervention.

	Patient *n* (%)	Organization * *n* (%)	System *n* (%)	Total *N* (%)
Screening	6 (22%)	16 (59%)	5 (19%)	27 (63%)
Diagnosis	0	0	4 (100%)	4 (9%)
Access to Treatment	2 (40%)	0	3 (60%)	5 (12%)
Access to Specialists	1 (33%)	1 (33%)	1 (33%)	3 (7%)
Adherence to Treatment	4 (100%)	0	0	4 (9%)
Retention in care	0	0	0	0

* No studies addressing only provider-level interventions were found. Several multicomponent organization-level interventions included provider-level elements.

**Table 2 ijerph-21-00529-t002:** Types of interventions by care continuum and level of intervention.

	Patient (Disease, *n*)	Organization (Disease, *n*)	System (Disease, *n*)
Screening	Decision aid (LC, 3) Education (LC, 3)	Automated phone reminder (DR, 1) Screening program (LC, 3) Patient navigator (LC, 1) QI with feedback (LC, 1) Comparison of screening methods (LC, 4) Telehealth (D, 6)	Awareness campaign (LC, 1) Mobile clinic (D, LC, 3)
Diagnosis	None	None	AI interpretation of imaging (D, 2) Mobile clinic (D, 1) Medicaid expansion (LC, 1)
Access to treatment	mHealth app (D, 2)	None	Multifaceted QI intervention (LC, 1) Medicaid expansion/insurance subsidy (MS, LC, 2)
Access to specialists	Telehealth (MS, 1)	Molecular tumor board (LC, 1)	Palliative care referrals (LC, 1)
Adherence to treatment	Multicomponent (MS, 2) Medication administration route (MS, 1) Companion presence (LC, 1)	None	None

MS = Multiple sclerosis; D = Diabetic retinopathy; LC = Lung cancer.

**Table 3 ijerph-21-00529-t003:** Interventions and main relevant findings for studies involving patients with multiple sclerosis.

Author	Brief Intervention Description	Main Relevant Findings
Baird, 2020 [21]	Behavioral intervention delivered in two phases over six weeks per phase. Phase one focused on sitting less; phase two focused on moving more.	The intervention was safe and feasible; there was a small positive change in sedentary behavior.
Cascione, 2018 [22]	Injectable versus oral disease-modifying therapies (DMTs)	At 48 weeks, there was higher adherence among those taking oral versus injectable DMTs.
Hartung, 2020 [23]	Low-income subsidy for Medicare beneficiaries newly diagnosed with MS	Across demographics, those who received a low-income subsidy were more likely to initiate early self-administered DMT than those who did not have a low-income subsidy.
Kinnett-Hopkins, 2018 [24]	Racially tailored exercise program for black persons with MS consisting of strength and aerobic activities, behavioral coaching materials, and supplemental content based on social cognitive theory	The intervention was feasible, effective, and safe; exercise behaviors increased in inactive participants.
Plow, 2019 [25]	A physical activity intervention versus a physical activity intervention plus a fatigue self-management intervention.	Fatigue management improved fatigue and quality of life at 12 weeks compared with social support, but not physical activity. Physical activity improved on quality of life compared with social support at 12 weeks.

**Table 4 ijerph-21-00529-t004:** Interventions and main relevant findings for studies involving patients with diabetic retinopathy.

Author	Brief Intervention Description	Main Relevant Findings
Abramoff, 2018 [26]	Trained site staff took images using nonmydriatic retinal camera and uploaded to the AI system	The AI system met the prespecified endpoints for superiority compared to the standard approach
Alam, 2019 [27]	Machine-learning approach to train and evaluate a model for AI classification of retinopathies	The study demonstrated that use of the AI model was feasible.
Al-Aswad, 2021 [28]	Mobile clinic with in-person evaluation; OCT and nonmydriatic fundus photography sent to an eye institute for analysis by ophthalmologist or optometrist in real time; videoconference conducted with patient to provide results	A small percent of patients screened positive for diabetic retinopathy and were referred for follow-up. Diabetic retinopathy was confirmed in most of those reached, and additional eye problems were detected in almost half.
Daskivich, 2017 [29]	Fundus photographs taken and uploaded by trained primary care clinic staff; off-site optometrist read photos to determine diabetic retinopathy grade, follow-up recommendations, and feedback; electronic results sent to PCP	Increased annual screening rates and reduced wait-time.
Hatef, 2017 [30]	Nonmydriatic fundoscopic camera to take retinal images at primary care visit, sent to an eye institute, evaluated, and returned to PCP. PCP recommended ophthalmologist follow-up for those with signs of diabetic retinopathy	Annual exam completion rate increased; a small percentage of those who had diabetic retinopathy identified in their scan and were referred to ophthalmologists completed the referral.
Jani, 2017 [31]	Patient’s retinal images taken by trained staff at a primary care visit and sent to a retinal specialist for remote review; specialist classified the level of diabetic retinopathy, gave recommendations, and sent results to PCP within 24 h	Post-implementation screening rate increased over pre-implementation screening rate. Some patients screened with diabetic retinopathy received a referral; some did not. Of those referred, the majority completed their referral visit within study period.
Liu, 2019 [32]	An existing teleophthalmology program that allows PCP to refer patients with walk-in scheduling	Patient barriers included unfamiliarity with teleophthalmology, misconceptions about screening, and logistical difficulties. Facilitators included PCP recommendation and convenience factors. PCP barriers included not knowing when screening was due and unfamiliarity with teleophthalmology. Facilitators included ease of referral process and communication of results.
Mehranbod, 2019 [33]	An automated telephone reminder a week prior to primary care appointment for screening in addition to a telephone reminder by medical assistant within a week of appointment	Attendance rates for appointments were lower among African American patients compared with Latino patients. Adding automated reminders improved attendance and narrowed the disparity in rates between African American and Latino patients; rates for both groups remain low.
Ramchandran, 2020 [34]	Patient care technician or nurse took digital photos of the eye; an ophthalmologist read images and uploaded disease and visual acuity results. Clinicians followed-up with patient based on reports in the EMR.	Patients rated teleophthalmology as highly as regular care, perceived high value of teleophthalmology, and were willing to pay an equivalent copay.
Rowe, 2021 [35]	Medical students provide ophthalmology screening services to patients under the supervision of one ophthalmology attending physician and one resident ophthalmologist.	The clinic showed significant cost savings for each screening conducted.
Serrano, 2018 [36]	Nurse took digital eye photos with dilation if deemed necessary; images uploaded to a website and reviewed by a fellowship-trained ophthalmologist at a university; patient returned for follow-up appointment with nurse to discuss results	Patients expressed satisfaction with telemedicine and preferred it to in-person visits; patients who had prior face-to-face exams were less likely to prefer telemedicine.
Tan, 2021 [37]	OCT obtained by eye doctors in a mobile van and interpreted remotely by retinal specialists.	Among patients with diabetes, OCT and clinical exam had moderate agreement in diagnosing retinopathy.

AI = artificial intelligence; OCT = optical coherence tomography; PCP = primary care practitioner; EMR = electronic medical record.

**Table 5 ijerph-21-00529-t005:** Interventions and main relevant findings for studies involving patients with lung cancer.

Author	Brief Intervention Description	Main Relevant Findings
Bagcivan, 2018 [38]	Early palliative care consultation (within 60 days of lung cancer diagnosis)	Early consultations addressed patient and family concerns not typically addressed in cancer care visits. Commonly evaluated symptoms were mood, general pain, and cognitive/mental status.
Beer, 2020 [39]	Breathe Easier app with mindfulness-based cancer recovery content	Primary benefits were convenience and having credible health information; top concerns included cost and difficulty of use.
Cardarelli, 2020 [40]	Application of Lung-RADS categories compared with retrospective LDCT results	Fewer additional tests using Lung-RADS compared with NLST. Among those with additional testing, the number identified with cancer was higher using Lung-RADS compared with NLST.
Cykert, 2019, 2020 [41,42]	Multi-faceted quality improvement intervention including a real-time warning system with missed appointments and deviations from standard timelines, quarterly clinical performance reports with aggregated completion of cancer treatments by patient race, nurse navigator, physician champion, and staff training on health equity	2019—Among patients in the intervention group, treatment completion rates did not differ between Black and White patients. Among patients in the control groups, Black patients had reduced treatment completion compared with White patients. 2020—Black and White patients in the intervention group had similar receipt of curative treatments. Black patients in the retrospective group had lower rates of receiving curative treatment compared with White patients.
Erkmen, 2021 [43]	Community engagement in churches and other community settings providing pamphlets with education and alliances with community leaders in each setting; CME for participating providers on screening program; lung cancer screening performed; paper and EMR forms for referrals to screening; SDM using a decision aid; radiology report provided by chest radiologist; imaging with LDCT; smoking cessation with pharmacology aids; 2-year follow-up by telephone	At 1 year, all people with Lung-RADS categories 3 or 4 adhered to follow-up screening, but only 23.7% of those with negative screens adhered. At 2 years, only 35.4% with positive screens and no cancer followed up.
Fung, 2018 [44]	A cancer prevention seminar providing Asian Americans with information about cancer prevalence and common cancers for Asian Americans, cancer risk factors and early warning signs of common cancers, cancer myths and facts, an overview of the American Cancer Society cancer screening guidelines, and actionable ways to reduce cancer risk.	Seminars developed for Cantonese-speaking Chinese Americans changed the beliefs and stated behaviors of Chinese Americans. Both groups had high knowledge at baseline. Changes in knowledge, attitudes, and screening intent were minimal between groups.
Huang, 2021 [45]	A molecular tumor board provided recommendations to clinicians for specific therapy and clinical trials based on patient diagnosis and next-generation sequencing testing results.	Compared to those with reviews, those without reviews had poorer survival.
Lau, 2021 [46]	A modified version of a web-based decision aid (shouldiscreen.com) with basic information about LDCT screening, education about lung cancer risk factors, and calculation of personalized lung cancer risk	Use of the decision aid led to small improvements in knowledge and increased concordance with current recommendations.
Li, 2020 [47]	AHRQ decision-aid “Is Lung Cancer Screening Right for Me?” translated in Chinese and adapted to health literacy and cultural needs	Participants reported that the adapted decision aid would facilitate informed decision making for LDCT screening. Based on reviewing the decision aid, the majority of patients understood causes and symptoms of lung cancer and LDCT screening and associated benefits, harms, and insurance coverage, although the majority were unable to understand the content without help.
Liu, 2020 [48]	State-level Medicaid expansion	Compared to men in states that did not expand Medicaid, those in states that did expand Medicaid had greater increases in 2-year survival and early-stage diagnosis. Outcomes for women did not differ among states that did and did not expand Medicaid.
Loehrer, 2018 [49]	State-level Medicaid expansion	Rates of complex surgical care increased relative to non-expansion states. The probability of undergoing surgical resection at high-volume hospitals did not change.
Olazagasti, 2021 [50]	NCCN screening criteria	Among patients already diagnosed with lung cancer, significantly more Hispanic/Latinx patients did not qualify for screening based on USPSTF guidelines compared with patients of other races. Rates of eligibility did not differ between African Americans and those who were White, Asian, or other races comparing the NCCN or USPSTF guidelines.
Otto, 2021 [51]	Presence of a companion at a patient care encounter with a medical oncologist	When a companion was present, oncologists provided more patient-centered communication and spent more time with patients. Oncologists perceived patients to be more active participants and to have more social support.
Owens, 2020 [52]	Breathe Easier app with content for mindfulness-based cancer recovery	The majority of participants thought the app was appropriate for African Americans, the information was well-understood, and that it would benefit lung cancer survivors to use the app. Participants were receptive to using the app but raised concerns of health literacy for others.
Pasquinelli, 2020 [53]	PLCOm2012 criteria	Among African American patients, the PLCO model had higher sensitivity for lung cancer screening compared with the UPSTSF guidelines.
Percac-Lima, 2018 [54]	Patient navigator support including brief smoking cessation counselling, reminding patients of CT screening, helping with translations, insurance issues, transportation concerns, other system barriers, and follow-up with patients about results from shared decision-making appointments with a primary care provider	The proportion of patients receiving CT screening via chest or lung CT was higher among those receiving the patient navigator compared with those receiving usual care.
Prosper, 2021 [55]	Use of LDCT to screen for lung cancer among at risk individuals	Among a synthesized sample of Black individuals, LDCT screening had a greater relative reduction in lung cancer mortality.
Raghavan, 2020 [56]	Mobile screening unit using a 35-foot coach with waiting area, portable LDCT scanner, high-speed wireless internet, and portable electronic tablet with smoking cessation education. Electronic images were sent for central review to an expert panel.	Screening identified 601 pulmonary nodules, including 267 participants with Lung-RADS 1, 183 participants with Lung-RADS 2, 62 participants with Lung-RADS 3, and 38 participants with Lung-RADS 4 lesions. Among those screened, 12 had lung cancer.
Randhawa, 2018 [57]	Free community LDCT screening program; tumor board review of Lung-RADS 3 or 4 findings and results sent to ordering physician by mail or via electronic records. Phone call to patients with results, and certified mail if needed	Screening identified 18.3%, 68.6%, 9.5%, and 3.6% of participants as Lung-RADS 1, 2, 3, and 4, respectively. Among physicians surveyed, 15% had never referred a patient. Barriers to referral included time constraints and precertification requirements.
Sender, 2019 [58]	Paper reminder placed in the chart for the provider to prompt screening prior to visit and providers received update when patients missed screenings.	After 6 months, screening rates improved compared with prior to the intervention.
Sferra, 2021 [59]	Option Grids brief information sheet to guide physician–patient encounters to discuss lung cancer screening options.	Patients randomized to Option Grids had lower decision regret and higher knowledge regarding next steps for positive screens and potential need for invasive procedures..
Springer, 2018 [60]	Campaign to increase lung cancer screening in rural Michigan using GoogleAds, gas station and convenience store flyers, and radio public service announcements.	Evidence did not show differences in screening rates between patients with more than and less than a 30-year smoking history.
Thurlapati, 2021 [61]	2018 NCCN Lung Cancer Screening Guidelines revised using an individualized risk-based Tammemagi Calculator to determine who should be screened for lung cancer	One-third of patients diagnosed with lung cancer did not meet the 2103 screening guidelines. Using the revised NCCN guidelines, 12.5% who were ineligible for screening would have been qualified for LDCT; however, 87.5% of those patients with lung cancer who were missed would still not have met screening criteria. Among those who did not meet screening guidelines, 50% were African American.
Townsend, 2021 [62]	LDCT screening program in a rural hospital setting following referral patterns in the 2013 USPSTF guidelines	Lung cancer was detected in 1.4% of screens over 8 years.
Williams, Looney, 2021 [63]	Four 90 min sessions provided by trained CHWs to educate community members about lung cancer screening and attitudes towards lung cancer.	Knowledge, perceived benefits of lung cancer screening, and self-efficacy increased and perceived barriers decreased among participants.
Williams, Shelton 2021 [64]	Trained CHW delivered a 90 min session with educational content including an overview of cancer screenings and risk, severity, benefits, and barriers to lung cancer screening and prevention.	The intervention helped reach more patients and educated them about cancer screenings. Participants improved on some, but not all, knowledge, benefit, and stigma measures.

Lung-RADS = Lung Imaging Reporting and Data Systems; NLST = National Lung Screening Trial; RCT = randomized controlled trial; LDCT = low-dose computed tomography; AHRQ = Agency for Healthcare Research and Quality; USPSTF = United States Preventive Services Task Force; NCCN = National Comprehensive Care Network; CME = continuing medical education; SDM = shared decision-making; EMR = electronic medical record.

## Data Availability

The original contributions presented in the study are included in the article/Appendix A, further inquiries can be directed to the corresponding authors.

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
