# Peer review of "A Rapid Review of Interventions to Improve Care for People Who Are Medically Underserved with Multiple Sclerosis, Diabetic Retinopathy, and Lung Cancer"

_ijerph, 2024, doi:10.3390/ijerph21050529_

Round 1
Reviewer 1 Report
Comments and Suggestions for Authors
This manuscript intends to analyse existing literature on interventions for medically underserved individuals with multiple sclerosis, diabetic retinopathy, and lung cancer in the United States. Authors utilize PRISMA protocols and search through PubMed and Web of Science. The manuscript reveals the importance of interventions across the care continuum, with a particular emphasis on screening. Evidence supports the introduction of telemedicine, mobile clinics, and insurance subsidies for a range of health issues.
I have read the paper with interest. Although the study reveals insights on the subject matter, I have concerns on the manuscript, which are listed below.
Comments:
· The manuscript does not provide justification for selection of two specific data bases for search. Why did the study not consider others such as Scopus, Google Scholar, etc?
· Authors should provide a convincing argument for the choice of last 5 years in their review. Why did they not consider last 10 years? 20 years?
· It may be useful to provide a table for keywords used in the search.
· Table 3 lies between page 6 and 17. Authors should find a better way to present this information. Dividing it into multiple tables, using figures, charts, etc.
· The study does not report regional (state, city) information for the reviewed articles. Are there regional variations in samples of studies?
· Social and economic determinants of health have significant role in health disparities. Are there any interventions studies addressing them for the specific sample groups considered by the manuscript?
· Numbering of references should be formatted with journal guidelines.
Reviewer 2 Report
Comments and Suggestions for Authors
The authors propose a rapid review of interventions to improve care for people who are medically underserved with multiple sclerosis, diabetic retinopathy, and lung cancer.
The introduction lacks a definition of the concept of medically underserved patients. Although the underlying concept of health disparities is presented, a clear definition of who are "medically underserved" should be given.
The numbering of the references in uncommon and not standardised across. the paper. It would be easier for the reader to have either the authors names or a unique number to refer to in the text and in Table3 to identify the reviewed papers.
The layout and content of the tables should be improved.
In Table 1, the percentage should be row-wise or column wise but not both (total margin are confusing).
Table 2 is difficult to read. The pullet points are everywhere. The use of superscript (not always) for the disease is noisy. Maybe using the first letter of the disease will help or extra small columns (MS | D | LC) with n the number of studies reviewed.
Table 3 should be presented in landscape format. It could be split per diseases and maybe sort/order by outcome or relevant findings rather than authors ? Different granularity in the information are given in the relevant findings' column which sometimes is too specific (ex . sensitivity/specifity details for the paper by Alam et al)A summary on what improvement it makes for medically underserved patients would be suffisant in such table (the whole picture could be in supplementary materials).
The results should be shorten. A lot of details are sometimes given for know observation or data presented in Table 3. Emphasize could be given to novelty like what is working or not working for medically underserved patients.
The discussion is well written.
Author Response
Please see attachment, thank you.

Reviewer 3 Report
Comments and Suggestions for Authors
This is a very interesting rapid review of medically underserved individuals with chronic conditions like MS, lung cancer, and diabetic retinopathy, and it is important. I am happy to see the detailed tables. Overall, I think it is well written. However, I have a few minor concerns/ recommendations for this paper that could improve it further:
1. What were the MESH terms and keywords? Usually they are listed in the methods section with Boolean operators
2. What is novel about this study?
3. Why were MS, lung cancer, and diabetic retinopathy selected as opposed to other types of chronic conditions? This is not clear.
4. Why are there numbered references in Roman numerals instead of Arabic numerals?
Thank you for the opportunity to review this manuscript.
Round 2
Reviewer 1 Report
Comments and Suggestions for Authors
Authors improved the paper although some comments are not addressed.
Reviewer 2 Report
Comments and Suggestions for Authors
The authors responded point to point to the first review with thoughful input which is greatly appreciated. The reading of the main manuscript has been improved. I have no further comment.